# Nutri-Score in the European Food Retail Supply: A Potential Incentive for Food Reformulation?

**DOI:** 10.3390/nu16234184

**Published:** 2024-12-03

**Authors:** Elly Steenbergen, Joline W. J. Beulens, Elisabeth H. M. Temme

**Affiliations:** 1Centre for Prevention, Lifestyle and Health, National Institute for Public Health and the Environment, 3721 MA Bilthoven, The Netherlands; liesbeth.temme@rivm.nl; 2Department of Epidemiology & Data Science, Amsterdam University Medical Centre, Location Vrije Universiteit, 1105 AZ Amsterdam, The Netherlands; j.beulens@amsterdamumc.nl; 3Amsterdam Public Health Research Institute, 1105 AZ Amsterdam, The Netherlands

**Keywords:** food reformulation, front-of-pack nutritional label, Nutri-Score, food composition, food retail supply, Europe

## Abstract

Background: To improve consumers’ diet, policy measures such as food reformulation strategies and front-of-pack nutritional labels (FOPNLs) are implemented, aiming to guide consumers’ food choice and to stimulate an improvement in food composition by manufacturers. The FOPNL Nutri-Score has been implemented in several European countries. Changes in food compositions in relation to the Nutri-Score over time have been limitedly studied. This study evaluates food compositions in Europe over time, and if changes in compositions of the food supply could have potentially resulted in changes in Nutri-Score classifications of foods. Materials and Methods: Food composition data were available from EUREMO, from which bread products, breakfast cereals, hot sauces, and processed potato products from Austria, Belgium, Finland, Italy, and the United Kingdom from 2019 to 2021 were selected (*n* = 2260). Of these countries, only Belgium had implemented the Nutri-Score in 2019. Distributions of food compositions and Nutri-Score classifications were calculated and changes in median salt, sugar, and saturated fatty acids content were plotted by food group, country and year. Distribution of the final sum of Nutri-Score points was plotted by nutrient, food group, country and year. Results: Overall, more favourable Nutri-Score classifications (i.e., towards Nutri-Score classification A) were observed in most of the selected food groups and countries over the years, due to the influence of specific nutrients such as salt in breakfast cereals (lower median of 0.1–0.4 g/100 g) and processed potato products (lower median of 0.1–1.2 g/100 g); and sugar in processed potato products (lower median of 0.1–1.9 g/100 g) and bread products (lower median of 0.7–2.2 g/100 g). For nutrient contents in other food groups, no consistent changes were observed. Conclusions: Changes in the compositions of the food supply resulted in favourable changes in Nutri-Score classifications, suggesting a potential for food reformulation. Monitoring after the actual implementation of Nutri-Score is recommended.

## 1. Introduction

Foods high in energy, salt, sugar and saturated fat contribute to unhealthy diets. Unhealthy diets in turn contribute to overweight and obesity, which are major preventable risk factors for non-communicable diseases such as type II diabetes and cardiovascular diseases. To improve consumers’ diet, the World Health Organization (WHO) has set guidelines to characterize healthy diets [1,2,3,4,5] as well as recommendations on food policy strategies to be implemented by governments to improve consumers’ dietary intake [6].

One food policy strategy is to encourage food manufacturers to reformulate foods to improve their nutritional composition. In Europe, national governments have implemented different reformulation strategies, often targeting the salt, sugar and saturated fat contents of foods by voluntary agreements with the industry to reduce contents or by setting upper limits by law [7,8,9]. Monitoring reformulation efforts is conducted by studying the compositions of branded foods. Several European studies and Joint Actions, such as Best practice in Reformulation, Marketing and Public Procurement (Best-ReMaP) [10] and Joint Action on Nutrition and Physical Activity (JANPA) [11], have collected branded food data in order to map and monitor the nutritional quality of foods to promote reformulation. The European Reformulation Monitoring study (EUREMO) was a study in which the salt, sugar and fat contents of branded foods were included and monitored in 16 European countries in order to provide tools for monitoring and assessing reformulation efforts [12].

Besides encouraging food reformulation, the WHO also recommends the implementation of a front-of-pack nutritional label (FOPNL) [13]. FOPNLs aim to guide consumers in making informed food choices. Meanwhile, FOPNLs also aim to stimulate food manufacturers to improve food compositions in order to apply a more favourable label on the packaging [14]. Several FOPNLs have been developed and implemented in Europe, such as the Heart symbol, the multiple traffic light system, and Nutri-Score. Nutri-Score was initially developed in France in 2017, as the official national FOPNL [15]. It has since been adopted in several other European countries, such as Belgium, Spain, Germany and the Netherlands [16]. An international governance structure, consisting of a steering committee, scientific committee and technical committee, assesses and updates Nutri-Score based on scientific evidence while incorporating and supporting requests and knowledge from stakeholders [17].

Nutri-Score is based on the Food Standards Agency Nutrient Profiling system in the United Kingdom which allocates points based on the food’s nutritional composition. Positive points are allocated to favourable components (i.e., protein, fibre and fruit, vegetables and legumes contents), whereas negative points are allocated to less favourable components (i.e., energy, salt, sugar and saturated fat contents). Three categories of Nutri-Score’s underlying algorithm (i.e., general foods, beverages, and fats, oils, nuts and seeds) calculate the Nutri-Score classification of a food, depending on the type of food, in a final sum of Nutri-Score points and its corresponding letter and colour, ranging from dark green A to dark orange E [18].

Although its implementation and alignment with food-based dietary guidelines has been debated, Nutri-Score was found to be successful in guiding consumers in their food choice in several European countries [19,20,21,22,23]. However, the ability of FOPNLs to encourage food manufacturers to improve foods has not yet been studied in depth [24] or only by scenario analyses. A previous modelling study in France found improved nutrient intakes by the population when substituting food purchases with more favourable alternatives based on Nutri-Score [25]. Previous studies from the Netherlands using data from before the actual implementation of Nutri-Score have shown that changes in the composition of foods result in a change towards a more favourable Nutri-Score classification. In a theoretical study, fixed changes in either the sodium, sugar or saturated fatty acids content resulted in potentially improved Nutri-Score classifications of foods for all foods groups except for legumes, using the original Nutri-Score algorithm from 2017 [22]. A recent Dutch study has shown actual changes in the nutrient compositions of branded foods between 2018 and 2020 while using the updated (2022) Nutri-Score algorithm, which resulted in changes towards more favourable Nutri-Score classifications for breakfast cereals, meat preserves, and sweets and sweet goods [26]. These results from the Netherlands indicate that Nutri-Score may be effective as an incentive for food manufacturers to stimulate food reformulation when implemented. However, only the Dutch retail supply was included.

These results need to be confirmed by data from other European countries to assess whether Nutri-Score could be a potential incentive for food reformulation for the European food retail supply. Therefore, the present study evaluates food compositions in several European countries over time and if changes in food composition would have resulted in changes in Nutri-Score classifications of foods.

## 2. Materials and Methods

### 2.1. Data Preparation

EUREMO data were used for the analyses, extraction date 7 February 2024. This dataset includes food compositions of food items categorized by food group and subgroup from several European countries (i.e., Austria, Belgium, Bulgaria, Denmark, Estonia, Finland, France, Greece, Hungary, Italy, Lithuania, Malta, Portugal, Romania, Slovenia, and the United Kingdom) from the years 2011 to 2022. Food composition data were extracted from images of food labels using applications of extraction software. Food groups covered 50% or more of the market share by country [12]. More detailed information on the background of EUREMO data and its data collection can be found elsewhere [12].

Figure 1 shows a flowchart of the selection procedure of foods included in the analyses. From the EUREMO dataset, data on Nutri-Score classifications were not available. Therefore, Nutri-Score classifications for foods had to be calculated. Food items were excluded when contents lacked the nutrients needed for the calculation of the Nutri-Score classification (i.e., energy, salt, saturated fatty acids, sugar, protein), except for fibre and the fruit, vegetables and legumes content (not mandatory on the back-of-pack) [27]. Additionally, the food groups fresh dairy products and desserts, and other products (heterogenous food group) were excluded as data were not sufficient to indicate and differentiate the type of food (e.g., drink or a solid food) as to whether the Nutri-Score classifications should be calculated using either the general, beverages or fats, oils, nuts and seeds algorithm (Nutri-Score guidelines state different calculation rules for solid foods, drinks or dairy products considered as added fats). Data from the years 2011 to 2018 only include foods sold in France and are of different food groups in each of these years. As these foods could not be compared over years, foods from France were excluded from the analyses (Table A1 in Appendix A). In 2022, only data from the UK and Italy were available and in a relatively small number, and were therefore excluded from the analyses. The majority of the data were from the years 2019 to 2021, in which a selection of food groups and countries was made for the analyses based on the total number of food items by food group (*n* > 1000), type of foods relevant to food reformulation (i.e., processed foods and foods of which the salt, sugar and saturated fatty acids content can be improved), and availability of food items in the years 2019 to 2021 by country (Table A2 in Appendix A). The food groups bread products, breakfast cereals, hot sauces and processed potato products from the countries Austria, Belgium, Finland, Italy and the United Kingdom were selected. Of the selected countries, Belgium is the only country that has implemented Nutri-Score (in 2019). Of the selected food groups, food group sizes smaller than 10 items in a specific country and year were excluded from the analyses.

For the analyses, assumptions have been made for the remaining missing values. For fibre content, the mean content of the subgroup was imputed for missing values. The fruit, vegetables and legumes content was assumed by subgroup for the calculation of the Nutri-Score (Table A3 in Appendix A).

### 2.2. Data Analyses

As there was no unique identifier available for the food items in the dataset to establish whether the same foods were available over the years, changes in compositions of these same foods within food groups over time could not be evaluated. Therefore, the number and type of items within subgroups of food groups were compared over the years (Table A3 in Appendix A). For this reason, descriptive results were provided to compare changes over time. The analyses involved the calculation of the distribution of nutrient contents by food group, country and year. For the salt, sugar and saturated fatty acids content (in g/100 g), changes in median contents over 2019–2021 were plotted by food group and country as these three nutrients are often targeted in reformulation strategies. Nutri-Score classifications were calculated for foods according to Nutri-Score guidelines [28] and their distributions were calculated by food group, country and year. The distributions of the final sum of Nutri-Score points were calculated and plotted by the salt, sugar and saturated fatty acids contents by food group, country and year.

All statistical analyses were performed in R software, version 4.4.1.

## 3. Results

An overview of the number of food items by selected food group, country and year is shown in Table 1. For Italy, data on bread products from only 2020 were included in the analyses. A total of 2260 foods were included in the analyses with group sizes ranging from 10 to 183 foods by food group, country and year (Table 1). Over the years, the proportion of subgroups within food groups changed (Table A3 in Appendix A). Pre-packaged breads within bread products increased over time in Austria (from 41/74 to 88/100) and the United Kingdom (from 17/25 to 46/56), whereas plain rusks decreased in Finland (from 101/183 to 24/179). The proportion of high-fibre cereals within breakfast cereals in Finland decreased over the years (from 22/67 to 4/22). Within hot sauces, numbers of tomato coulis and similar increased in Austria (from 0/31 to 10/38) and Italy (from 3/24 to 19/55) over time, whereas pesto sauces, sauces with tomatoes and cheese, sweet and sour sauces and other sauces from around the world decreased to zero in Finland in 2020. The proportion of French fries for the deep fryer within processed potato products increased in all selected countries over the years (Table A3 in Appendix A).

Table A4 in Appendix A shows the distributions of the nutrient contents of each of the selected food groups, countries and years. Differences in the median salt, sugar and saturated fatty acids contents of food groups by country between 2019 and 2021 are shown in Figure 2. Lower median contents were observed for each of the three nutrients as well as unchanged and higher median nutrient contents in the most recent year. Over time, bread products had an unchanged median salt content in all countries, except for Austria. Bread products contained less sugar in all countries (median difference of 0.7–2.2 g/100 g). In addition, bread products in Finland contained less saturated fatty acids over the years. For breakfast cereals, the median salt content was lower with 0.1–0.4 g/100 g in all countries over the years, whereas the sugar content was only lower in Austria (Table A4 in Appendix A and Figure 2). In Austria, breakfast cereals improved in all three selected nutrients as the three median nutrient contents were lower in the most recent year. Hot sauces in all countries, except for Finland, had a lower median salt content (median difference of 0.1–0.6 g/100 g) by 2021. Except for the United Kingdom, the median sugar content of hot sauces was also lower (0.2–1.1 g/100 g) over the years. Italy was the only country with a lower median saturated fatty acids content of hot sauces over time (Table A4 in Appendix A and Figure 2). Processed potato products have improved in median salt (median difference of 0.1–1.2 g/100 g) and sugar (median difference of 0.1–1.9 g/100 g) content in all countries over the years. Additionally, processed potato products from Belgium and Finland had lower median contents for all three nutrients. Italy was the only country in which one of the median nutrient contents was higher (i.e., median saturated fatty acids content increased by 0.3 g/100 g) (Table A4 in Appendix A and Figure 2).

Figure 3a,b and Table A5 in Appendix A show the distributions of calculated Nutri-Score classifications of food groups by country and year. The distributions show a shift towards a more favourable Nutri-Score classification (i.e., towards Nutri-Score classification A) in most of the selected food groups and countries over the years. In Belgium and Finland, bread products shifted towards Nutri-Score classification A through the years (2 and 10 percentage points increase in Nutri-Score classification A, respectively), whereas in Austria and the United Kingdom Nutri-Score classifications unfavourably shifted towards C (16 percentage points increase) and B (9 percentage points increase), respectively. For breakfast cereals, Nutri-Score classifications improved towards A in Austria, Belgium, Italy and the United Kingdom over the years (10, 1, 5 and 9 percentage points increase for Nutri-Score classification A). Breakfast cereals in Finland changed unfavourably towards Nutri-Score classification C and D (19 and 11 percentage points increase). Nutri-Score classifications of hot sauces shifted towards A, especially in Austria (sum of 21 percentage points increase for Nutri-Score classifications A and B), Finland (6 percentage points increase for Nutri-Score classification B), Italy (sum of 18 percentage points increase for Nutri-Score classifications A and B) and the United Kingdom (1 percentage point increase for Nutri-Score classification A). Processed potato products improved towards Nutri-Score classifications A over the years in all countries (sum ranging from 7 to 73 percentage points increase for Nutri-Score classifications A and B). Distributions of Nutri-Score classifications ranged over three to five classifications, except for bread products in Italy in 2020 and processed potato products in Italy and the United Kingdom in 2019. For these latter, Nutri-Score classifications range over two classifications (i.e., C and D).

Figure 4a–c show the final sum of Nutri-Score points plotted by the salt, sugar and saturated fatty acids contents (respectively) of foods by food group, country and year. These figures show the influence of the specific nutrient alone on the final sum of Nutri-Score points. The influence of the salt content on the final sum of Nutri-Score points of foods is most visible for hot sauces and processed potato products as foods with lower salt content have a lower final sum of Nutri-Score points (i.e., towards Nutri-Score classification A) and foods with higher salt content have a higher final sum of points (Figure 4a). Foods from the most recent year were also found to be more on the left bottom side of the plots than foods from other years of some food groups in countries, showing that foods from more recent years are generally lower in salt content as shown in Figure 3. This can be observed for breakfast cereals in Austria and the United Kingdom, hot sauces in Austria, and processed potato products in Austria, Belgium, Finland and the United Kingdom (Figure 4a).

The influence of the sugar content on the final sum of Nutri-Score points was most visible for bread products and breakfast cereals (Figure 4b), whereas the influence of the saturated fatty acids content of foods was visible in all food groups (Figure 4c). Foods from the most recent year were visibly lower in sugar content and the final sum of Nutri-Score points for breakfast cereals in Austria, hot sauces in Finland and processed potato products in Austria (Figure 3 and Figure 4b). The saturated fatty acids content was lower in the most recent year, especially for breakfast cereals in Austria and processed potato products in Finland (Figure 3 and Figure 4c).

## 4. Discussion

The present study aimed to assess whether Nutri-Score could be a potential incentive to food manufacturers for food reformulation in the European food retail supply, by evaluating food compositions in European countries over time and if changes in compositions of the food retail supply could have resulted in changes in Nutri-Score classifications. Overall, most changes in the median salt, sugar and saturated fatty acids content and Nutri-Score classifications of foods were favourable as observed in the selected food groups and countries from 2019 to 2021. Some unfavourable changes were observed as well, such as the higher sugar content in breakfast cereals in all countries except for Austria. Depending on the food group, specific nutrients influenced the Nutri-Score classification as lower contents resulted in a more favourable Nutri-Score classification (e.g., salt in breakfast cereals and processed potato products) and vice versa.

The results in the present study show that changes in the food supply as well as its food composition have taken place from 2019 to 2021 in several European countries and food groups. These findings are in line with what was previously found in the Netherlands [26], confirming the indications that Nutri-Score classifications could be improved by favourably changing the nutrient composition of foods. As for the other way around, Nutri-Score may act as a disincentive to food manufacturers to worsen the composition of foods.

The changes in food composition may reflect food reformulation by food manufacturers as each of the countries in the present study have implemented reformulation strategies targeting either the salt, sugar or saturated fat content of foods before or during the years 2019 to 2021 [7,8]. One may define food reformulation as the change in food composition of the same food item over time, but food reformulation is also defined as the change in the composition of the food retail supply. This does not necessarily mean the change in food composition of an individual food item, but new food items could also have been introduced that have more favourable compositions [29]. In the present study, it was not possible to distinguish food reformulation as to whether it resulted from the improvement in existing foods or by the introduction of new foods, as a unique identifier of foods in the dataset, such as the European Article Number, was not available. However, the number of food items by subgroup differed for some food groups in several countries over the years. Changes in the composition of the food group may therefore be due to a change in the proportion of subgroups within a subgroup, and not necessarily due to the reformulation of foods. In the present study, food reformulation reflects both the change in the composition of existing food items as well as the change in the composition of the food retail supply due to introductions to or eliminations of food items from the market. This was also found in a previous study from the Netherlands, in which changes in nutrient contents were found to be due to the improvement in existing foods as well as due to the introduction of new food items within the food groups [26]. In Belgium, before Nutri-Score was officially implemented, breakfast cereals had already improved in mean salt (−20%), sugar (−5.2%) and saturated fatty acids (−4%) content. These improvements were found in individual foods that were available in both 2017 and 2018 [30]. The present study showed comparable improvements in the salt content of breakfast cereals in Belgium from 2019 to 2020. However, the sugar and saturated fatty acids content increased during these years (20.5 to 22.3 g and 0.9 to 1.3 g, respectively). Reformulation of breakfast cereals occurred to existing foods and prior to the actual implementation of Nutri-Score in Belgium from 2017 to 2018 [30].

Nutri-Score has currently been implemented in several European countries, but not in all of the countries or during the years that were studied in the present study. Austria and Italy have not officially implemented, or announced the intention to implement Nutri-Score, whereas since 2000 Finland has implemented the Heart symbol [31] and the United Kingdom has had its multiple traffic light system since 2013 [32]. In Belgium, where Nutri-Score has been implemented since 2019 [33], the implementation of Nutri-Score has impacted efforts by food manufacturers to improve food compositions of breakfast cereals as observed in a previous study [30]. Studies on the effects of FOPNLs on food reformulation by food manufacturers are still limited. A study suggested improved food compositions of store-brand cakes by household purchases in Great Britain after the implementation of the multiple traffic light system [24]. Other FOPNLs in countries may have also had an impact on the observed food compositions since foods in Europe are regularly exported to other countries. Additionally, national reformulation strategies during these years may have resulted in the improvements that were found in the food compositions. As Nutri-Score was only implemented in Belgium during the years studied, the present study does not show the efforts that food manufacturers might put into food reformulation when Nutri-Score is actually implemented. Besides, Nutri-Score classifications were calculated for all foods available of the selected food groups. In practice, Nutri-Score may not be applied to all foods as it is still a voluntary food policy measure. Food manufacturers may choose not to apply Nutri-Score on their foods, for example when they only manufacture foods that will receive unfavourable Nutri-Score classifications. A mandatory approach on the use of FOPNLs may have more impact as suggested before [34]. Several national food reformulation strategies are also voluntary [7] and may therefore not incentivize all food manufacturers to improve their foods.

Although Belgium was the only country to have implemented Nutri-Score during the years and of the countries studied, the present study shows that nutrient compositions and calculated Nutri-Score classifications have improved for the selected food groups, countries and years. Moreover, changes in the contents of specific nutrients seemed to be reflected by Nutri-Score classifications and their change over the years. This suggests that Nutri-Score could potentially incentivize food manufacturers to improve food compositions. These efforts may show even more when monitoring food compositions in countries which have already implemented the Nutri-Score. Therefore, it would be highly recommended to monitor the nutrient compositions and Nutri-Score classifications of foods after the actual implementation of Nutri-Score. Monitoring compositions of the food retail supply and Nutri-Score classifications of foods after its implementation will validate whether Nutri-Score is able to incentivize food manufacturers to improve foods, especially when looking at foods that are available in several years.

The present study used EUREMO data, which consist of food composition data of several food groups from several countries and years. As composition data were not (sufficiently) available for all food groups, countries and years, a selection was made for the analyses in this study. In addition, the dataset did not provide sufficient information to establish and determine the Nutri-Score algorithm to be used to calculate classifications for fresh dairy products and desserts according to Nutri-Score guidelines. For the analyses regarding Nutri-Score, the fruit, vegetables and legumes content of foods was not available and therefore assumed, as well as the missing fibre content of foods for which the mean fibre content was imputed for missing contents by subgroup. Calculated Nutri-Score classifications may therefore deviate from actual values. Additionally, in this dataset, foods were already categorized into food groups, which combined a minimum of 50% of the market share. Therefore, not the whole food retail supply was included in the dataset, even though the weighted selection aimed to establish a sample representative of actual purchases and consumption [12]. The dataset did also not provide sufficient information to identify and establish whether foods were available in multiple years, although the type of foods were known within food groups over the years. Yet, comparisons of identical foods over years were not possible. As the data collection of foods within food groups was based on market share, the proportion of types of foods available within food groups in countries and years differed for some of the food groups. This limitation may have affected the results regarding the changes (both favourable and unfavourable) that were observed for food compositions and calculated Nutri-Score classifications, although most of these changes represented the actual food retail supply. Furthermore, some changes were rather large for the relatively short period of time. This could be due to changes in market shares of food items and in combination with smaller food group sizes. These changes should also be interpreted cautiously. The EUREMO dataset has its strengths as it includes data on branded food items, rather than generic food items, and from multiple European countries over multiple years. Using food composition data of foods from different brands, food groups and different countries provides valuable insights into the composition of the European food retail supply.

## 5. Conclusions

The present study showed favourable changes in the median salt, sugar, and saturated fatty acids content in several of the selected food groups (bread products, breakfast cereals, hot sauces and processed potato products) and countries (Austria, Belgium, Finland, Italy and the UK) from 2019 to 2021. With these changes, in the case of Nutri-Score classifications being applied on all foods under study, Nutri-Score classifications would have shifted towards more favourable classifications. These findings suggest that Nutri-Score could be a potential incentive for food manufacturers to improve food compositions. To establish whether and to what extent food manufacturers are incentivized by Nutri-Score to improve foods, monitoring of nutrient composition and Nutri-Score classifications of foods over years, especially after Nutri-Score is implemented, is recommended.

## Figures and Tables

**Figure 1 nutrients-16-04184-f001:**
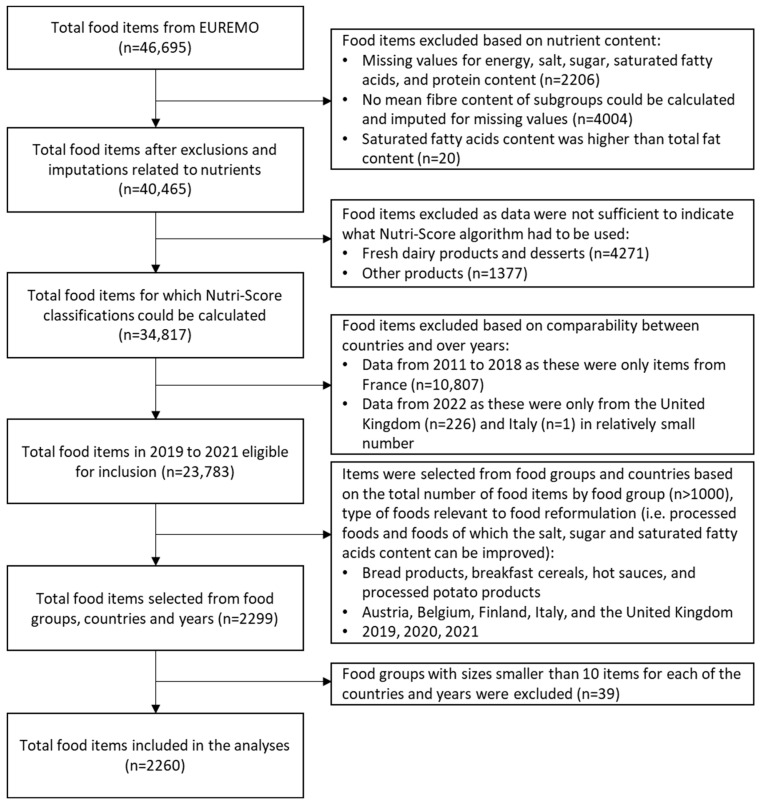
Flow chart of the selection of food items included in the analyses.

**Figure 2 nutrients-16-04184-f002:**
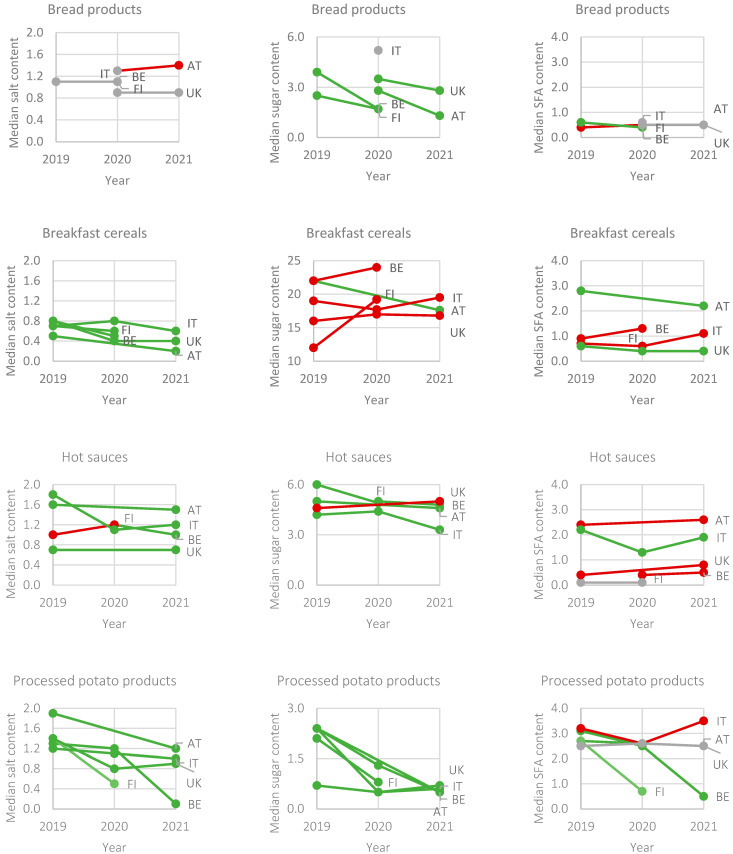
Changes in median salt, sugar and saturated fatty acids (SFAs) contents (g/100 g) by food group and country over years 2019 to 2021. Green lines indicate decreased nutrient contents, red lines indicate increased nutrient contents and grey lines indicate unchanged nutrient contents. AT = Austria, BE = Belgium, FI = Finland, IT = Italy, UK = United Kingdom. See Table A4 in Appendix A for the distributions of nutrient contents by food group, country and year.

**Figure 3 nutrients-16-04184-f003:**
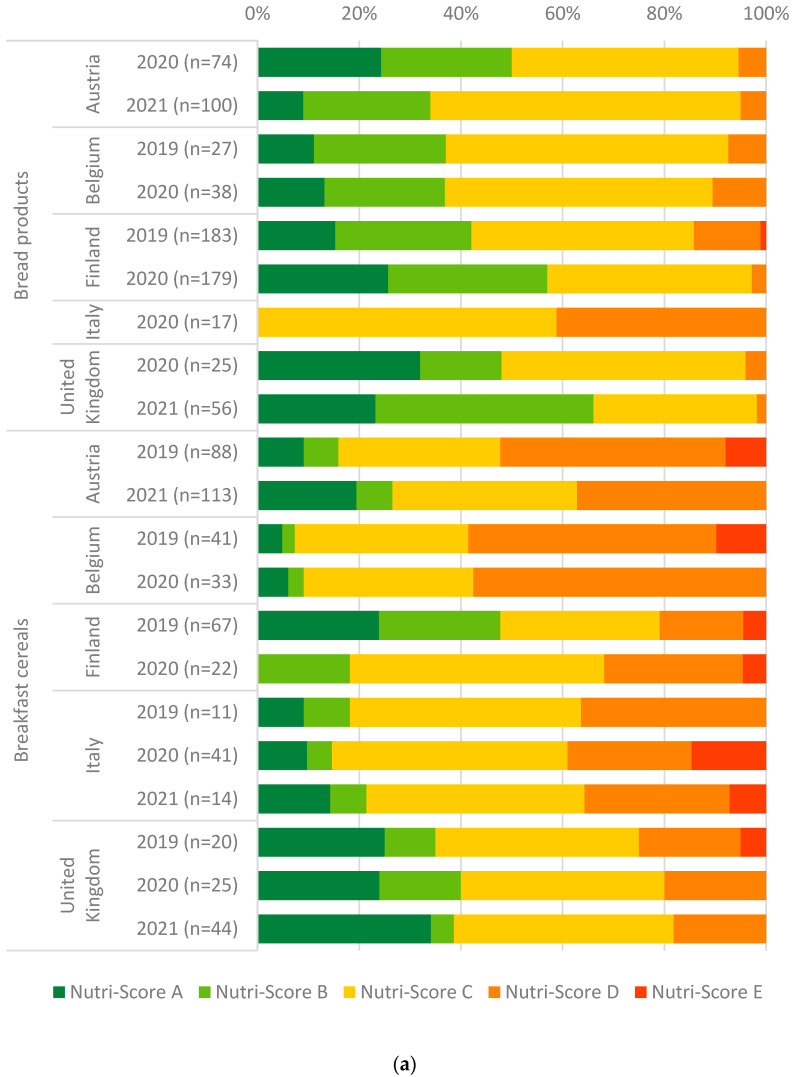
(**a**). Distribution of Nutri-Score classifications calculated for bread products and breakfast cereals by country and year. (**b**). Distribution of Nutri-Score classifications calculated for hot sauces and processed potato products by country and year.

**Figure 4 nutrients-16-04184-f004:**
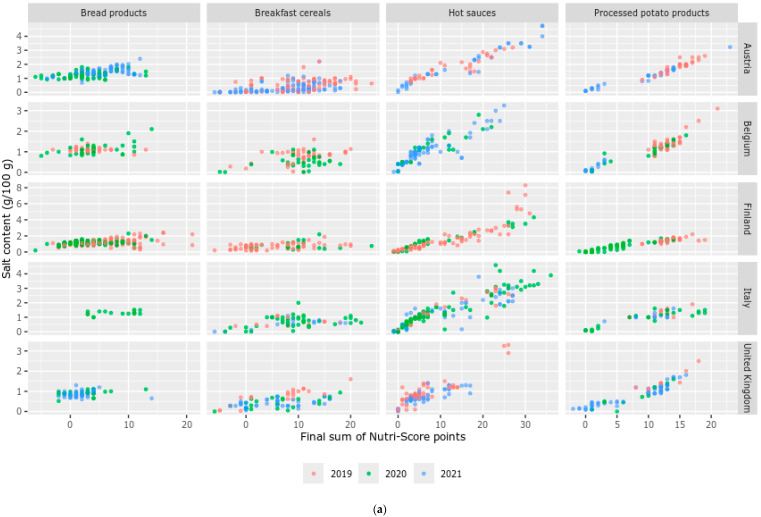
(**a**). Salt content (g/100 g) plotted by the final sum of Nutri-Score points by food group and country from 2019 to 2021. (**b**). Sugar content (g/100 g) plotted by the final sum of Nutri-Score points by food group and country from 2019 to 2021. (**c**) Saturated fatty acids content (g/100 g) plotted by the final sum of Nutri-Score points by food group and country from 2019 to 2021.

**Table 1 nutrients-16-04184-t001:** Overview of the number of food items by selected food group, country and year.

Food Group	Country	N in 2019	N in 2020	N in 2021
Bread products	Austria	0	74	100
	Belgium	27	38	0
	Finland	183	179	0
	Italy	9	17	9
	United Kingdom	9	25	56
Breakfast cereals	Austria	88	9	113
	Belgium	41	33	1
	Finland	67	22	0
	Italy	11	41	14
	United Kingdom	20	25	44
Hot sauces	Austria	31	2	38
	Belgium	0	25	85
	Finland	80	35	0
	Italy	24	94	55
	United Kingdom	74	0	66
Processed potato products	Austria	31	0	39
	Belgium	39	40	22
	Finland	47	85	0
	Italy	10	25	15
	United Kingdom	10	13	59

N = number of food items. Grey cells indicate the food groups of a specific country and year in which the group size is smaller than 10 items. These were not included in the analyses of this study.

## Data Availability

The Food and Beverages Labels Explorer (FABLE) is an information system developed by the Joint Research Centre (JRC) of the European Commission. Collection of data hosted in FABLE, and used for this research, was supported with funding from the European Union’s Health Programme (2014–2020) through the EUREMO project. The data are available on request via https://food-labels-explorer.jrc.ec.europa.eu/en, accessed on 25 November 2024.

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
