# Peer review of "Nutri-Score in the European Food Retail Supply: A Potential Incentive for Food Reformulation?"

_nutrients, 2024, doi:10.3390/nu16234184_

Round 1

Reviewer 1 Report

Comments and Suggestions for Authors

The paper described improvement of foods over years because of reformulation. Authors present this as an achievement of NS, but that is incorrect: this study does not support NS. Rather the other way round: improvements in composition following reformulation leads to a better NS value. As such any (all…) nutrient profiling system could/should be associated with improved values. This stance should be clearly repositioned. If this is not repositioned the paper is incorrect.

Hence authors should not position this paper in support of NS but inversely: food reformulation leads to improvements in (all…) nutrient profiling system. Moreover, NS should be deleted from the title and mentioned only scarcely along other NP systems in the abstract and text.

Belgium is the only country in this survey that has introduced NS. So why run a study on countries that do not have NS …with NS ?? It is not valid to take such data to support one NP system, i.e. NS (that is not even practiced in most of the investigated countries).

From figures 2 and 3 I see values going up and down in just 1 or 2 year trajectories. Food reformulation cannot go that fast. So there must be bias (selection bias) in these data. Moreover, I see changes going up the one year and down the next.

Belgium: Line 21 = 2019; line 131 = 2018 ??

I note that the authors are from NL. NL, PL and IT have many scientists opposing the NS system. This should be acknowledged in the paper and properly referenced.  

“Nutritional composition” and similar wording is better presented as “nutrient composition”. This applies to many places in the text.

Author Response

Response to Reviewer 1 Comments

1. Summary

Thank you very much for taking the time to review this manuscript. We agree largely with your comments and believe we share the same view on how to phrase and present the findings towards the reader. Please find the detailed responses below and the corresponding revisions/corrections highlighted/in track changes in the re-submitted files.

2. Questions for General Evaluation

Reviewer’s Evaluation

Does the introduction provide sufficient background and include all relevant references?

Must be improved

Is the research design appropriate?

Can be improved

Are the methods adequately described?

Yes

Are the results clearly presented?

Must be improved

Are the conclusions supported by the results?

Must be improved

3. Point-by-point response to Comments and Suggestions for Authors

Comment 1: The paper described improvement of foods over years because of reformulation. Authors present this as an achievement of NS, but that is incorrect: this study does not support NS. Rather the other way round: improvements in composition following reformulation leads to a better NS value. As such any (all…) nutrient profiling system could/should be associated with improved values. This stance should be clearly repositioned. If this is not repositioned the paper is incorrect. Hence authors should not position this paper in support of NS but inversely: food reformulation leads to improvements in (all…) nutrient profiling system. Moreover, NS should be deleted from the title and mentioned only scarcely along other NP systems in the abstract and text.

Response 1: Thank you for pointing this out. We agree largely with this comment. The aim of this study, as described in the Introduction (p. 2-3, lines 97-99), was “to evaluate the food compositions in several European countries over time and if changes in food composition would have resulted in changes in Nutri-Score classifications of foods”. Hence, we have looked at and observed that, overall, favourable changes in food compositions also resulted in favourable changes in Nutri-Score classifications. The aim of this study was not to support nor oppose the implementation of Nutri-Score itself, but to explore Nutri-Score’s second aim, which is to incentivize food manufacturers to improve their foods. Our study is not limited nor generalized to all existing front-of-pack nutritional labels (FOPNLs), but studied Nutri-Score’s potential specifically, since it has been officially implemented in the Netherlands. In addition, no mention was done regarding food reformulation as an achievement of Nutri-Score since we studied, indeed, the other way around: changes observed in the food compositions over time would have resulted in changes in Nutri-Score classifications, in the case when Nutri-Score would have been applied to the foods. To clarify and address these points, we revised the Introduction by adding examples of other existing FOPNLs in Europe (p. 2, lines 59-60) and by adding references and a sentence on previous findings from France suggesting that food reformulation may increase improved nutrient intakes by the population (p. 2 lines 78-82). As Nutri-Score is one of the main outcomes of this study and other FOPNLs have not been studied in our study, we believe that Nutri-Score should be kept as the main subject of this article and be included in its title.

Comment 2: Belgium is the only country in this survey that has introduced NS. So why run a study on countries that do not have NS …with NS ?? It is not valid to take such data to support one NP system, i.e. NS (that is not even practiced in most of the investigated countries).

Response 2: Thank you for pointing this out. We were indeed not able to obtain and use data from all countries and years in which Nutri-Score was implemented due to availability, as described in Materials and methods 2.1 Data preparation (p. 3, lines 110-135). As for the purpose of this study, it did not aim to monitor the use and changes of actually applied Nutri-Score classifications, but rather aimed to evaluate – with the data available – whether Nutri-Score could be a potential incentive for food manufacturers if changes in food composition were shown to be able to change Nutri-Score classifications. This has also been described in the Discussion, with recommendations for further research to validate our findings (p. 16, lines 343-348). As Nutri-Score was implemented in Belgium during the years that were studied, we elaborated on the discussion regarding the observations from Belgium in this study (p. 15, lines 316-319). We also revised the sentences in the Discussion to elaborate on the implementation of Nutri-Score and other FOPNLs in countries that have (potentially) impacted the countries’ food compositions (p. 16, lines 319-324). Additionally, we added “… data on Nutri-Score classifications were not available. Therefore, Nutri-Score classifications for foods had to be calculated.” (p. 3, lines 111-112) and “Of the selected countries, …” (p. 3, lines 132-133) to make it more clear that Nutri-Score classifications were calculated for the foods and that only Belgium had actually implemented Nutri-Score.

Comment 3: From figures 2 and 3 I see values going up and down in just 1 or 2 year trajectories. Food reformulation cannot go that fast. So there must be bias (selection bias) in these data. Moreover, I see changes going up the one year and down the next.

Belgium: Line 21 = 2019; line 131 = 2018 ??

Response 3: Thank you for your comment. We agree with the observation that both favourable and unfavourable changes in food compositions were observed in the study (as mentioned in the Results, second and third paragraph). As described in the Discussion (p. 15, lines 284-303), food reformulation is not necessarily defined as the improvement of the individual foods but it may also include the introduction of new foods with more favourable compositions. In addition, the present study showed differences in proportions of food types in some countries over the years, meaning that some types of foods within a food group were more present than other types of foods in different years. This could also have occurred due to the method of the data collection by the EUREMO study. For each food group, data were collected that covered a minimum of 50% of the market share. Changes in market shares of foods may be representative for actual purchases and consumption, but may also have had impact on the changes observed for the compositions of that food group in this study. We have clarified this by revising the sentences in the last paragraph of the Discussion (p. 16-17, lines 350-370). We also added sentences to address the point regarding the favourable changes observed in a short period of time (p. 17, lines 370-373): “Besides, some changes were rather large for the relatively short period of time. This could be due to changes in market shares of food items and in combination with smaller food group sizes. These changes should also be interpreted cautiously.” The typo in line 134 on page 3 should be “2019” and has been updated in the manuscript.

Comment 4: I note that the authors are from NL. NL, PL and IT have many scientists opposing the NS system. This should be acknowledged in the paper and properly referenced.

Response 4: The aim of this study was to explore Nutri-Score’s second aim (which is to incentivize food manufacturers to improve their foods) by evaluating the composition of the food retail supply over time in the case of applying Nutri-Score classifications to the foods. This exploration and evaluation of the food retail supply in Europe with respect to improving Nutri-Score classifications is a first step towards establishing whether Nutri-Score could be an incentive for food manufacturers to improve their foods. This study provided recommendations in order to further study Nutri-Score’s potential, and stands separate from the discussion by supporters and opponents regarding the implementation of Nutri-Score itself. We have revised the Introduction and briefly addressed it by adding “Although its implementation and alignment with food-based dietary guidelines has been debated, …” (p. 2, lines 76-77), but do not go into further detail regarding supporting and opposing scientists of Nutri-Score in this study.

Comment 5: “Nutritional composition” and similar wording is better presented as “nutrient composition”. This applies to many places in the text.

Response 5: Thank you for pointing this out. We have revised the wording when applied to the current study throughout the manuscript.

4. Response to Comments on the Quality of English Language

No comments on the quality of English language

5. Additional clarifications

We revised some wording and structuring of sentences throughout the manuscript.

We have elaborated the Data Availability Statement (p. 17, lines 395-398).

We would like to emphasize that it is not the study’s purpose to support or oppose Nutri-Score as a policy tool to be implemented, nor to study its performance compared to other existing FOPNLs. Rather, as Nutri-Score has already been officially implemented in the Netherlands (as well as in some other countries in Europe), this study aimed to explore Nutri-Score’s purpose and potential in incentivizing food manufacturers by evaluating food compositions and calculated Nutri-Score classifications, in the case of Nutri-Score being applied to foods.

Reviewer 2 Report

Comments and Suggestions for Authors

I enjoyed very much reading this article, because it emphasizes very well the usefulness of Nutri-Score.

There are some improvements recommended , in order to improve it:

- you briefly mentions other front-of-pack (FOP) labeling systems, such as the Heart Symbol in Finland and the Traffic Light system in the UK. I recommend to delve a little bit deeply into their comparative effectiveness or interactions with Nutri-Score. Without this discussion, there is a limiting of a holistic understanding of how Nutri-Score performs relative to alternative labeling systems. Why is Nutri-Score better?! I am sure you know there are many discussions regarding Nutri as being more useful than other FOP. Some countries being deeply against Nutri-Score

- while your article notes some unfavorable trends (e.g., increased sugar content in certain products), the discussion does not fully explore why these occurred or the limitations of Nutri-Score in preventing such outcomes. Maybe elaborate a bit in this area.

- you also generalize a little bit, and do not take intro consideration specific contexts. the discussion could have explored in more depth why some countries or food groups saw greater improvements than others. For instance, cultural differences, market structures, or existing food policies could have been analyzed further. Please, do not elaborate a lot, but this kind of topic is necessary for the articvle. 

Author Response

Response to Reviewer 2 Comments

1. Summary

2. Questions for General Evaluation

Reviewer’s Evaluation

Does the introduction provide sufficient background and include all relevant references?

Yes

Is the research design appropriate?

Yes

Are the methods adequately described?

Yes

Are the results clearly presented?

Yes

Are the conclusions supported by the results?

Yes

3. Point-by-point response to Comments and Suggestions for Authors

Comments 1: You briefly mentions other front-of-pack (FOP) labeling systems, such as the Heart Symbol in Finland and the Traffic Light system in the UK. I recommend to delve a little bit deeply into their comparative effectiveness or interactions with Nutri-Score. Without this discussion, there is a limiting of a holistic understanding of how Nutri-Score performs relative to alternative labeling systems. Why is Nutri-Score better?! I am sure you know there are many discussions regarding Nutri as being more useful than other FOP. Some countries being deeply against Nutri-Score.

Response 1: Thank you for pointing this out. We conducted this study to evaluate whether changes in food compositions could also have led to changes in Nutri-Score classifications. This has been previously observed in the Netherlands, but needed to be confirmed with data from other European countries, as it is of interest when implementing Nutri-Score. As the Netherlands has implemented Nutri-Score since the beginning of 2024, this study is an exploration of Nutri-Score’s potential as an incentive for food reformulation, rather than a substantiation of performing ‘better’ than other existing front-of-pack nutritional labels (FOPNLs). As other FOPNLs do exist and are being applied to foods in Europe, we have included them in the Discussion (p. 15-16, lines 313-317). As to your comment, we have elaborated the point by adding findings from a previous study from Great Britain (p. 16, lines 320-322) and that foods – carrying Nutri-Score or another FOPNL) may have impact on food compositions in other countries due to export (p. 16, lines 322-324).

Comment 2: While your article notes some unfavorable trends (e.g., increased sugar content in certain products), the discussion does not fully explore why these occurred or the limitations of Nutri-Score in preventing such outcomes. Maybe elaborate a bit in this area.

Response 2: Thank you for you comment. We have revised and updated the last paragraph of the Discussion (p. 16-17, lines 350-370) to emphasize that the data collection (which was based on the market shares of foods by the EUREMO study) may have resulted in differences in the proportion of types of foods over the years to which changes in food composition and calculated Nutri-Score classifications may have been affected. We have also added a sentence addressing the trends that were observed: “Besides, some changes were rather large for the relatively short period of time. This could be due to changes in market shares of food items and in combination with smaller food group sizes. These changes should also be interpreted cautiously.” (p. 17, lines 370-373).

Comment 3: You also generalize a little bit, and do not take intro consideration specific contexts. the discussion could have explored in more depth why some countries or food groups saw greater improvements than others. For instance, cultural differences, market structures, or existing food policies could have been analyzed further. Please, do not elaborate a lot, but this kind of topic is necessary for the articvle.

Response 3: Thank you for you comment. We revised and added sentences in the Discussion (p. 16, lines 316-324) regarding the implementation of Nutri-Score and other FOPNLs potentially affecting food compositions within countries as well as the export of foods carrying FOPNLs. We also updated the last paragraph of the Discussion (p. 16-17, lines 365-370) in which we emphasized that data collection was based on market share and that changes in market share and proportion of types of foods over the years (even though representing purchase and consumption by consumers) may have affected the food compositions and calculated Nutri-Score classifications, favourably or unfavourably.

4. Response to Comments on the Quality of English Language

No comments on the quality of English language

5. Additional clarifications

We revised some wording and structuring of sentences throughout the manuscript.

We have elaborated the Data Availability Statement (p. 17, lines 395-398).

We would like to emphasize that it is not the study’s purpose to support or oppose Nutri-Score as a policy tool to be implemented, nor to study its performance compared to other existing FOPNLs. Rather, as Nutri-Score has already been officially implemented in the Netherlands (as well as in some other countries in Europe), this study aimed to explore Nutri-Score’s purpose and potential in incentivizing food manufacturers by evaluating food compositions and calculated Nutri-Score classifications, in the case of Nutri-Score being applied to foods.

Round 2

Reviewer 2 Report

Comments and Suggestions for Authors

Thank you for answering to our request, the article has improved.